# Long-Term Results after Autologous Breast Reconstruction with DIEP versus PAP Flaps Based on Quality of Life and Aesthetic Outcome Analysis

**DOI:** 10.3390/jcm12030737

**Published:** 2023-01-17

**Authors:** Angela Augustin, Evi M. Morandi, Selina Winkelmann, Ines Schoberleitner, Daniel Egle, Magdalena Ritter, Thomas Bauer, Tanja Wachter, Dolores Wolfram

**Affiliations:** 1Department of Plastic, Reconstructive and Aesthetic Surgery, Medical University of Innsbruck, Anichstrasse 35, 6020 Innsbruck, Austria; 2Department of Obstetrics and Gynecology, Medical University of Innsbruck, Anichstrasse 35, 6020 Innsbruck, Austria

**Keywords:** breast cancer, mastectomy, autologous breast reconstruction, PAP flap, DIEP flap, donor site morbidity, quality of life, aesthetic outcome, scar quality, Breast-Q

## Abstract

(1) Background: This work aimed to conduct a comparative study, providing long-term data about patient-reported outcome measures and donor site scar assessments, as well as an aesthetic evaluation of the reconstructed breasts in patients with DIEP versus PAP flap breast reconstruction. (2) Methods: This prospective, single-center, matched cohort study included a total of 36 patients after DIEP and PAP flap breast reconstruction. The evaluation was carried out using the Breast-Q and POSAS questionnaire, as well as the Breast Aesthetic Scale for cosmetic analysis, by four plastic surgeons. (3) Results: The postoperative Breast-Q evaluation revealed no significant differences between both patient groups for the categories of the physical well-being of the donor site, the physical well-being of the breast, and satisfaction with the breast. A scar evaluation of the donor site region showed equivalent results for the thigh and abdomen regions, concerning the overall opinion of the patients and the observers. There was no significant difference between both methods of reconstruction for all aspects of breast aesthetics. (4) Conclusions: Similar results for donor site morbidity, scar quality, and the aesthetic outcome of the breasts in both the DIEP and PAP patient groups have been demonstrated. Hence, in those cases suitable for both types of reconstruction, the decision can be based on factors such as patients’ lifestyles, leisure activities, and preferences.

## 1. Introduction

Breast Cancer is the leading cancer entity in women, with a lifetime risk of approximately 12.9% in industrialized countries [1]. Increasing survival rates over the last four decades changed and diversified the treatment challenges, with aesthetic demands and life quality claims being of increasing importance [1,2]. Despite significant therapeutic advances, NSME and SSME still remain an inherent part of the therapeutic options and have an even broadened ambit, with bilateral and prophylactic mastectomies rising in numbers [3,4]. Women clearly benefit from postmastectomy breast reconstruction in their long-term survivorship period [5]. Among the different methods of reconstruction, autologous techniques have proven their superiority due to their permanent and natural aesthetic results and high patient satisfaction when compared to implant-based techniques [5,6,7,8]. Free flap reconstruction with tissue from the abdomen has long been favored and the deep inferior epigastric perforator (DIEP) flap is nowadays considered the gold standard [9,10]. However, the thigh has proven to be a safe alternative donor site; more precisely, the profunda artery perforator (PAP) flap has evolved to be a reliable option, especially for the reconstruction of small- to medium-sized breasts [11,12]. Since microsurgical safety has been proven for both perforator flaps, considerations about the long-term results, donor site morbidity, and recovery time come to the fore. This necessitates a detailed work-up of the autologous techniques to help surgeons in their decision-making process and, as a further consequence, to permit thorough patient counseling. In order to allow informed consent, all reconstructive options must be well discussed. Moreover, the quality of the provided information in the counseling process is crucial for postoperative patient satisfaction [13]. A literature review revealed three comparative studies about patient-reported donor site morbidity in autologous perforator flap breast reconstruction, with diverging patient preferences being reported [14,15,16]. This work aimed to conduct a comparative study providing long-term results about patient-reported outcome measures and donor site scar assessment, as well as an aesthetic evaluation of the reconstructed breasts in patients with DIEP versus PAP flap autologous breast reconstruction.

## 2. Materials and Methods

### 2.1. Study Design

This prospective, single-center, cohort study of 36 patients after autologous breast reconstruction was approved by the Institutional Ethics Committee of the MEDICAL UNIVERSITY INNSBRUCK (protocol code 1058/2020, 28 October 2020). Two matched patient groups of 18 patients each (after PAP and DIEP flap reconstruction) were evaluated. 

Informed consent for photo documentation, the operation, and the anonymized evaluation and publication of data was obtained in written form from all patients. Inclusion criteria were defined as age > 18 years, breast cancer diagnosis, high-risk genetic disposition or recurrent infections of the breast, uni- or bilateral breast reconstruction, and postoperative course longer than 12 months. We excluded patients with metastatic disease, severe psychiatric disorders, and a follow-up of less than 12 months. Patient demographics, including age, body mass index (BMI), smoking habits, and comorbidities, as well as postoperative complications, were documented retrospectively. 

A prospective evaluation of both patient groups was carried out using two validated questionnaires; a translation to German language was conducted in accordance with the copyright owners. Patient-reported outcome and quality of life were assessed using the postoperative Breast-Q version 2.0 (Copyright ©2017, Memorial Sloan Kettering Cancer Center and the University of British Columbia). An evaluation of the questionnaire was carried out using the provided conversion tables. The scores ranged from 0 to 100; the higher the scores, the more favorable the results. Patients with missing responses to specific questions were removed from the analysis of the related question but were kept for the analysis of completed questions.

Evaluation of the scar quality in the donor site region was carried out with POSAS version 2.0 (The Patient and Observer Scar Assessment Scale, https://www.posas.nl/, accessed on 20 August 2022) [17]. The patient and the observer scale contain 6 items each, and all of them are rated using a 10-point score. The lowest score is ‘1’, which corresponds to the situation of normal skin, while a score of 10 equals the largest difference from normal skin. Additionally, for each scale, there is a rating for the overall opinion about the scar.

Cosmetic evaluation of the breast reconstruction for both patient groups was conducted by a gender-balanced panel of four independent plastic surgeons (two seniors and two residents) based on standardized postoperative photo documentation in frontal and oblique views using a German translation of the Breast Aesthetic Scale [18]. Nine questions of this validated tool were used to evaluate the aesthetic result of the breast reconstruction, and questions about the nipple-areola reconstruction were omitted since this was not the aim of this study. Every question is graded from 1 to 5, with 5 points representing the perfect aesthetic result. In the DIEP group, only 17 patients were included for aesthetic evaluation since one patient did not complete photo documentation. 

### 2.2. Patients

All patients with a PAP flap breast reconstruction from January 2016 to November 2019 in our department were identified. In this period, a total of 29 patients underwent autologous breast reconstruction with a PAP flap. Six patients were excluded due to metastatic disease or severe psychiatric disorders, and three patients were lost to follow-up due to relocation. A total of 20 patients with a post-op follow-up of at least 12 months were invited for clinical examination, of whom 18 patients consented to participate, and nine of those underwent bilateral PAP flap breast reconstruction. Consequently, 27 flaps and 27 donor sites were evaluated. The evaluation data of the PAP cohort was recently published by our group [19]. Consequently, our department’s patient database was searched retrospectively from January 2011 to December 2020, and 18 patients, after DIEP flap breast reconstruction, were matched with this PAP cohort according to age, clinical diagnosis, and concomitant oncologic therapy as the criteria. All except one patient were matched according to laterality as well. In both groups, one patient with recurrent infections was included, but only the patient in the DIEP group had bilateral reconstruction. Therefore, in the DIEP group, 28 flaps and 18 donor sites were evaluated. All included patients gave informed consent and were enrolled for clinical examination, photo documentation, and prospective questionnaire evaluation. The mean follow-up in the DIEP group was 69.8 (±34.7) months (Figure 1a). The mean postoperative follow-up in the PAP group was 34.0 (±15.8) months (Figure 1b).

### 2.3. Statistical Analysis

Statistical analysis was performed using ©Microsoft Excel 2016 (Microsoft Corporation. https://office.microsoft.com/excel; accessed 20 August 2022) and ©MedCalc Statistical Software Ltd. (MedCalc Software Ltd. Fisher exact probability calculator. https://www.medcalc.org/calc/fisher.php, version 20.115; accessed 27 September 2022). Chi-square or Fisher exact tests and Student t-tests were used to test for differences between the PAP versus DIEP groups; statistical significance was set at a p-value less than 0.05 for all tests.

## 3. Results

### 3.1. Patient Characteristics

In both groups, 18 patients were evaluated after autologous breast reconstruction with PAP and DIEP flaps, respectively. Bilateral reconstruction was performed in the PAP group in 9 patients (*n* = 27 flaps) and, in the DIEP group, in 10 patients (*n* = 28 flaps). The patient information is listed in Table 1. Matching of the patient groups was carried out based on age, indication for mastectomy, and adjuvant oncologic therapies; therefore, there were no significant differences between the PAP versus DIEP group in terms of radiation and chemotherapy. The average BMI was significantly higher in the DIEP patients (25.3 kg/m^2^) compared to the PAP patients (21.6 kg/m^2^; *p* = 0.001). Additionally, flap volume (565.2 ± 207.4 vs. 327.7 ± 108.2; *p* < 0.0001) and mastectomy volume (518.1 ± 167.3 vs. 274.8 ± 132.8; *p* < 0.0001) were significantly higher in DIEP patients. The mean length of postoperative hospital stay was 9.6 (±2.9) days in the DIEP group and 10.8 (±4.4) days in the PAP group (*p* = 0.3675).

Postoperative complications were defined as events classified as Grade 3b according to the Clavien–Dindo Classification, necessitating operative revision under general anesthesia [20]. Such events (hematoma, seroma, wound dehiscence, and wound infection) occurred in 29.6% (8/27) of operated thighs and 5.5% (1/18) of abdominal donor sites (*p* = 0.0479). One flap loss was observed in each group due to venous failure. Evaluation of postoperative complications in the breast, as well as rates of secondary corrections in the breast or donor site, did not reveal significant differences between both patient groups; the detailed results are shown in Table 2. 

### 3.2. Breast-Q

All included patients completed the Breast-Q form at least 12 months postoperatively. Patients who underwent bilateral surgery answered one questionnaire for each reconstructed breast and, in the PAP group, for each donor site also. The results are shown in Figure 2. The postoperative Breast-Q evaluation revealed no significant differences between both patient groups in the categories of physical well-being donor site (*p* = 0.9478), physical well-being breast (*p* = 0.1840), and satisfaction with the breast (*p* = 0.0745) (Figure 2a). The section about satisfaction regarding the donor site contains three questions, namely about the appearance of the scar, the position of anatomic landmarks (umbilicus/gluteal crease), and how the donor site looks without clothes. The rating was carried out using a score from 1 (very unsatisfied) to 4 (very satisfied). A comparison of both patient groups did not reveal significant differences regarding donor site satisfaction. The scores are shown in Figure 2b. An evaluation of the categories, psychosocial well-being (*p* = 0.0031) and sexual well-being (*p* = 0.0469), revealed significantly higher Q-scores in the DIEP group (Figure 2a).

### 3.3. POSAS

All included patients in this study completed the POSAS questionnaire for donor site scar evaluation at least 12 months postoperatively. All parameters were scored from 1 to 10, with the lowest score 1 corresponding to a normal skin situation; the results are given in Table 3.

The patients’ rating regarding the overall opinion on their donor site scars indicated a mean value of 3.67 (±2.08) in the DIEP group and 4.72 (±2.41) in the PAP group (*p* = 0.1453) and therefore did not show significant differences (Figure 3a). The observers’ rating regarding the overall opinion on the donor site scars resulted in a mean value of 2.72 (±1.41) in the DIEP patients and 3.16 (±1.29) in the PAP patients (*p* = 0.3074), not showing differences between both groups either (Figure 3b). However, in both patient groups, the overall opinion about the donor site scar showed improved ratings in the observers’ assessment compared to the patients’ results, with a significant difference in the PAP group (*p* = 0.0068) (Table 3, Figure 3c,d).

The POSAS score is calculated by summing up the scores of all six items, excluding the overall opinion. The patient form asks about pain, itching, color, stiffness, thickness, and irregularities; the observer form asks about vascularity, pigmentation, thickness, relief, pliability, and surface area. A comparison between the DIEP patient group (19.72 ± 8 points) and the PAP patient group (20.78 ± 9.94 points) did not reveal any differences concerning the total score (*p* = 0.7198). Still, the DIEP group reported significantly more itching from the scar than the PAP group (2.39 ± 1.77 points versus 1.26 ± 0.84 points; *p* = 0.0076).

A comparison of the two study groups also showed equivalent results concerning the total observer score (*p* = 0.2329). However, scar relief was significantly better rated in the DIEP group (2.28 ± 1.52 points versus 3.4 ± 1.41 points in the PAP group; *p* = 0.0196).

### 3.4. Cosmetic Results

A comparison of the two study groups showed equivalent results concerning the overall opinion about the aesthetic outcome, with a mean of 3.41 points in the PAP group and 3.61 points in the DIEP group (*p* = 0.4771). Likewise, there was no significant difference between both methods of reconstruction for all other aspects of breast aesthetics; the details are given in Table 4.

## 4. Discussion

The rising number of patients seeking autologous breast reconstruction after a mastectomy leads to a steadily growing experience with a variety of surgical options [21]. Patients need to be evaluated and counseled thoroughly to allow for a distinct selection process for the individual patient’s best choice. The abdominal DIEP flap is now considered the gold standard for perforator flap reconstruction, but also the thigh-based PAP flap has proven to be a safe alternative [9,11,22,23]. The PAP flap is particularly favored in patients with a rather low BMI and lack of abdominal tissue and in patients with previous abdominal surgery, potentially compromising perforator vessels [10]. A comparative evaluation of the two perforator flaps is necessary to support the process of decision-making in patients suitable for both options.

In this study, matching the patient groups was performed according to age and indication for mastectomy and adjuvant therapy, with the aim of minimizing the bias on the aesthetic outcome and the patient’s perception of their illness. The patient groups did significantly differ concerning their average BMI, with a higher BMI in the DIEP group. The average BMI in the PAP group was 21.6 kg/m^2^; these patients often did not provide sufficient abdominal fat for reconstruction. Furthermore, feasible reconstruction volume is also limited regarding thigh-based reconstructions since the PAP flap presents a lower average flap volume [16,24]. This often limits PAP reconstruction to small- up to medium-sized breasts. These known disadvantages do also correspond to our findings of a significantly lower flap volume and lower mastectomy volume in the PAP cohort [12]. Besides flap characteristics, donor site morbidity must be well considered for an evaluation of the surgical methods. Preliminary studies demonstrate that functional deficits are rare in thigh donor sites, but they may cause concern regarding the ability to sit for prolonged periods or the restriction of leisure activities, such as bike-riding, stretching, hiking, and climbing [25]. Haddock reports a return to normal lower extremity musculoskeletal condition after six months and high patient satisfaction after PAP harvesting [23]. More literature is available about the long-term morbidity of the abdominal DIEP donor site, where abdominal wall hernia and bulge occur in 2–7% of cases, and abdominal weakness represents the leading functional impairment in a wide range of 7–64% [26,27,28,29,30,31,32]. In this study, postoperative donor site complications that necessitated operative revision were found significantly more often in the PAP group; this is inconsistent with the recent literature, where comparable incidences of surgical site complications in DIEP and PAP donor sites are described [14,24,33]. We observe that, with increasing experience and familiarity with the PAP flap, the complication rates in thighs are decreasing, which is the case in our department, which might hint at the learning curve as the main cause. An evaluation showed that secondary corrections to the donor site due to aesthetic or functional deficits were more frequent in the DIEP group, where 27.8% of patients had staged touch-up procedures, compared to 11.1% in the PAP group, although this difference was not significant. Murphy et al. described superior patient satisfaction with the PAP donor site in their comparative study [16]. We hypothesize that patients may put higher aesthetic demands on the well-visible abdominal donor sites and, therefore, DIEP patients might be more willing to undergo surgical corrections. We believe that information about the potential need for follow-up procedures should be part of the initial patient counseling in order to build up adequate expectations about the final result (Figure 4).

Despite these differences in the postoperative and follow-up period concerning complications and secondary corrections, the Breast-Q evaluation in our study showed that a comparison between both patient groups did not reveal significant differences regarding donor site satisfaction and the physical well-being of the donor site. Notably, the significantly higher postoperative complication rate in the PAP group does not compromise long-term patient satisfaction and quality of life concerning the donor site region. A comparative evaluation of donor site satisfaction in the literature reports diverging results so far. Lee reports similar levels of satisfaction for the abdomen and the thigh, whereas Murphy found a superiority of the PAP donor site, and Haddock’s patients had a preference for the DIEP donor site [14,15,16]. Furthermore, a Breast-Q evaluation of the categories of physical well-being and satisfaction with the breast revealed no significant differences between both patient groups. However, a postoperative Breast-Q evaluation revealed significantly higher Q-scores in the DIEP group for the categories of psychosocial well-being and sexual well-being. We hypothesize that this may be biased due to our study schedule; the evaluation of the DIEP patients was carried out in 2022, whereas the PAP patients were questioned in 2020, coinciding with the first COVID wave in Austria. During this time, the population faced long periods of governmentally dictated home quarantine. The impact of this psychologically exceptional situation must be considered since a decrease in people’s quality of life (with the poorest levels among females) has been reported in conjunction with the pandemic’s consequences [34]. An inferiority in psycho-social well-being in the PAP group may therefore not essentially be linked to the technique of breast reconstruction.

The POSAS evaluation showed that the patient’s, as well as the observer’s, ratings regarding the overall opinion on the donor site scars did not show significant differences between the abdomen and the thigh. Itching was the only scar symptom experienced significantly more often in the abdomen compared to the thigh. So far, two studies using the POSAS have been completed for DIEP scar evaluation after breast reconstruction. The POSAS scores were inferior in both studies compared to our results, but Siegwart reports a better patient and observer overall opinion about scar quality in the DIEP group without a mesh [13,35]. Patient information about donor site scar outcome is crucial to maximizing postoperative scar appraisal [13]. Some patients might have the preconception that flap harvesting is equivalent to the widely known body-contouring procedures since similar surgical incisions are made, but the aesthetic outcome after abdominoplasty has been described as superior to the abdominal donor site after autologous breast reconstruction [36]. Such diverging presumptions might also contribute to our findings of a better donor site scar rating among the observers compared to patients in the POSAS evaluation (Figure 3c,d). Surgeons know about the challenges of flap harvesting for autologous breast reconstruction. In body contouring procedures, surgical resection is limited to the excess skin and fat. On the contrary, especially in thin patients, sufficient flap volume for autologous breast reconstruction can only be gained by extensive tissue harvesting. As a result, the closure of the donor site is sometimes only achieved under tension, even using wide mobilization. This represents a major difference between reconstructive surgery and the information that must be provided beforehand to patients. In our department, we use pre and postoperative photographs from selected cases in the first counseling session in order to build up expectations to be as realistic as possible regarding the results. Still, our data show that these challenges are independent of the choice of the donor site.

In this study, there were no differences between the DIEP and the PAP patients in the long-term aesthetic outcome of the breasts. Although flap volume was significantly lower in our PAP patients, both of the patient groups showed equivalent ratings for breast reconstruction, being proportional to their body habitus. However, selection bias is evident since PAP reconstruction was mainly performed on thin patients. We believe that the equality of the PAP and DIEP flap, concerning the aesthetic outcome, is also stressed by the rate of the staged, secondary corrections of the breast, where our evaluation showed comparable numbers in both patient groups.

There are several limitations to our study. The single-institution design allows only insight into the results from our department. Reconstructive surgery was performed by three senior surgeons in the PAP group; in the DIEP group, two extra senior surgeons were involved. Therefore, differing surgeon-related outcomes cannot be excluded completely. Another limitation might be represented by the patient sample size; however, a clear strength of the study is the matching of the patients, creating homogeneity between both groups included in the study.

## 5. Conclusions

This study provides extensive information on the long-term results after autologous perforator flap breast reconstruction. Similar results for donor site morbidity, scar quality, and the aesthetic outcomes of the breast in both the DIEP and PAP patient groups have been demonstrated. In cases suitable for both types of reconstruction, this allows patients to choose a method of reconstruction based on their lifestyle, leisure activities, and preferences. Our data will support surgeons and patients in the complex decision-making process of breast reconstruction.

## Figures and Tables

**Figure 1 jcm-12-00737-f001:**
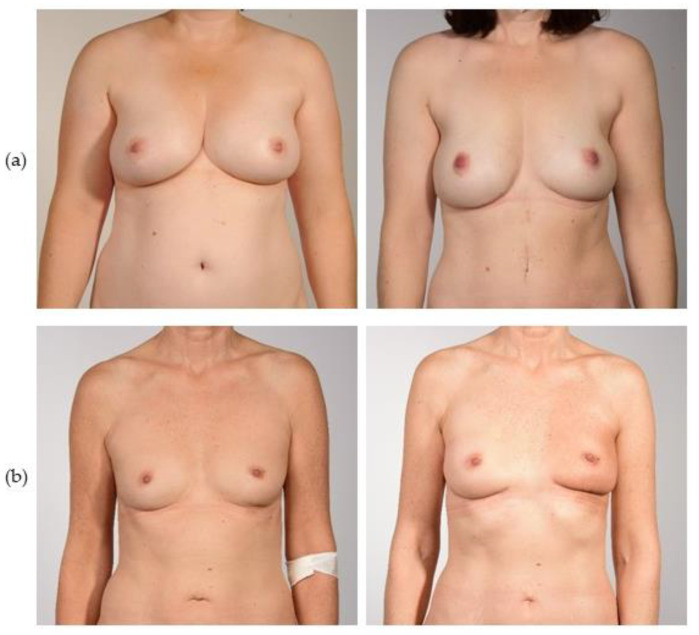
(**Left**) Preoperative and (**right**) postoperative views of bilateral autologous breast reconstruction with (**a**) DIEP and (**b**) PAP flaps.

**Figure 2 jcm-12-00737-f002:**
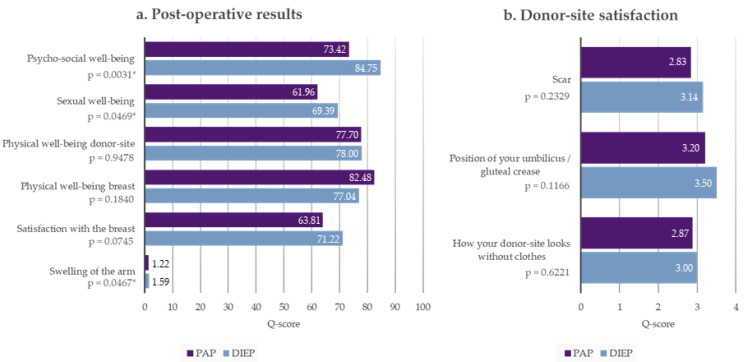
Breast-Q evaluation. (**a**) Postoperative results; (**b**) donor site satisfaction. * *p* < 0.05.

**Figure 3 jcm-12-00737-f003:**
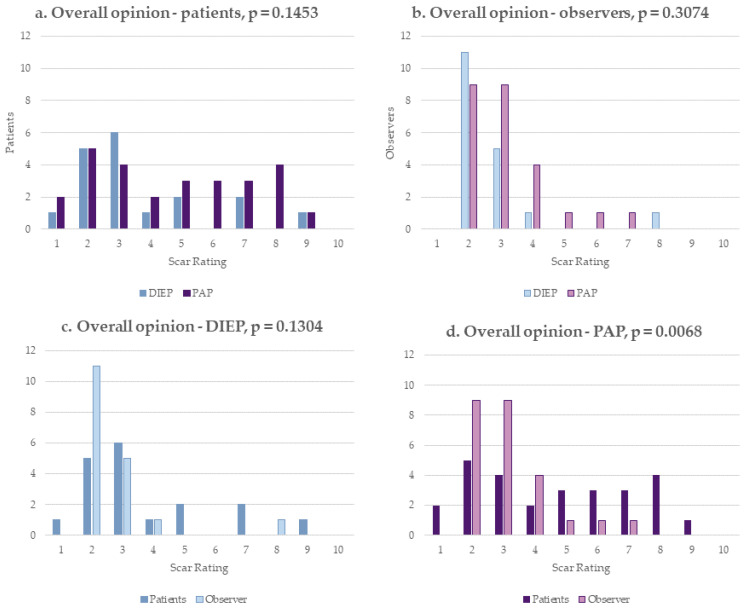
POSAS–overall opinion on donor site scar. (**a**) Patient scale; (**b**) observer scale; (**c**) DIEP cohort; (**d**) PAP cohort.

**Figure 4 jcm-12-00737-f004:**
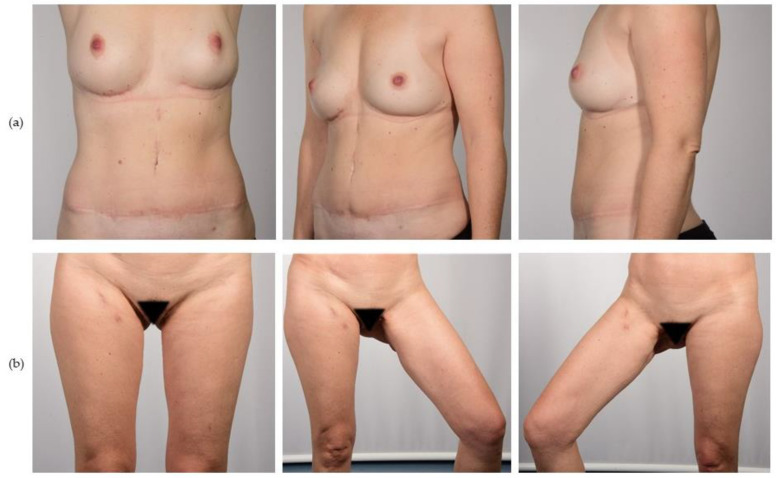
Postoperative views of the donor sites of (**a**) DIEP and (**b**) bilateral PAP flap.

**Table 1 jcm-12-00737-t001:** Patient characteristics.

Characteristic	DIEP		PAP		*p*-Value
	Mean	(±std)	Mean	(±std)	
Age (years) *	41.3	(±6.7)	43.6	(±7.4)	0.3660
Follow-up (months) *	69.8	(±34.7)	34.0	(±15.8)	0.0005
BMI (kg/m^2^) *	25.3	(±3.7)	21.6	(±2.3)	0.0010
Flap volume (cc) †	565.2	(±207.4)	327.7	(±108.2)	<0.0001
Mastectomy volume (cc) †	519.1	(±167.3)	274.8	(±132.8)	<0.0001
Postoperative hospital stay (days)	9.6	(±2.9)	10.8	(±4.4)	0.3675
	** *n* **	**(%)**	** *n* **	**(%)**	
Active smoker *	4	(22.2)	1	(5.6)	0.3377
Time of reconstruction †					0.0515
Primary	28	(100)	23	(85.2)	
Secondary	0	(0.0)	4	(14.8)	
Indication for mastectomy †					0.4182
Breast cancer	14	(50.0)	17	(63.0)	
Nonmalignant	14	(50.0)	10	(37.0)	
Prophylactic	12	(42.9)	9	(33.3)	
Mastitis	2	(7.1)	1	(3.7)	
Positive genetic testing *	4	(22.2)	4	(22.2)	>0.99
Radiotherapy *					>0.99
Yes	9	(50.0)	10	(55.6)	
Previous radiotherapy	1	(5.6)	3	(16.7)	
Adjuvant	8	(44.4)	7	(38.9)	
No	9	(50.0)	8	(44.4)	
Chemotherapy *					
Yes	12	(66.7)	11	(61.1)	>0.99
Previous chemotherapy	9	(50.0)	7	(38.9)	
Adjuvant	3	(16.7)	4	(22.2)	
No	6	(33.3)	7	(38.9)	

100%: * *n* = patients, † *n* = flaps.

**Table 2 jcm-12-00737-t002:** Post-operative complications (Clavien–Dindo 3b) and secondary corrections.

Characteristic	DIEP *n*	(%) *	PAP *n*	(%) †	*p*-Value
Complications breast	4	(14.3)	6	(22.2)	0.5027
Complications at donor site	1	(5.5) ^x^	8	(29.6)	0.0479
Secondary corrections breast	5	(18.5) ^≈^	9	(34.6) ^~^	0.2238
Secondary corrections donor site	5	(27.8) ^x^	3	(11.1)	0.2351

100%: * *n* = 28 DIEP flaps; ^x^
*n* = 18 DIEP donor sites; ^≈^
*n* = 27 DIEP flaps (1 flap loss). † *n* = 27 PAP flaps, ^~^
*n* = 26 PAP flaps (1 flap loss).

**Table 3 jcm-12-00737-t003:** POSAS–patient and observer scale of donor site scar.

POSAS Score	DIEP		PAP		
Patient Scale	Mean	(±std)	Mean	(±std)	*p*-Value
Has the scar been painful for the past few weeks?	1.33	(±1.15)	1.93	(±1.88)	0.2498
Has the scar been itching for the past few weeks?	2.39	(±1.77)	1.26	(±0.84)	0.0076
Is the scar color different from the color of your normal skin?	4.00	(±1.83)	4.52	(±2.44)	0.4556
Is the stiffness of the scar different from your normal skin?	4.06	(±2.55)	4.52	(±2.56)	0.5635
Is the thickness of the scar different from your normal skin?	3.94	(±2.63)	4.19	(±2.40)	0.7585
Is the scar more irregular than your normal skin?	4.00	(±2.67)	4.37	(±2.74)	0.6627
Total score	19.72	(±8.50)	20.78	(±9.94)	0.7198
**Overall opinion**	**3.67**	**(±2.08)**	**4.72**	**(±2.41)**	**0.1453**
**Observer scale**					
Vascularity	2.39	(±1.64)	2.42	(±1.11)	0.9495
Pigmentation	3.11	(±1.63)	3.80	(±1.72)	0.2032
Thickness	2.33	(±1.49)	3.08	(±1.09)	0.0714
Relief	2.28	(±1.52)	3.40	(±1.41)	0.0196
Pliability	3.00	(±1.70)	2.36	(±0.79)	0.1150
Surface area	2.89	(±1.97)	3.12	(±1.39)	0.6623
Total score	15.16	(±9.30)	18.08	(±6.32)	0.2329
**Overall opinion**	**2.72**	**(±1.41)**	**3.16**	**(±1.29)**	**0.3074**

**Table 4 jcm-12-00737-t004:** Breast aesthetic scale.

Questions	DIEP		PAP		
Breast	mean	(±std)	Mean	(±std)	*p*-Value
Breast symmetry	3.22	(±0.92)	3.21	(±1.14)	0.9733
Breast position	3.66	(±0.75)	3.67	(±1.01)	0.9496
Inframammary fold	3.82	(±0.8)	3.68	(±0.98)	0.5954
Volume	3.58	(±0.75)	3.51	(±1.09)	0.7995
Shape and contour	3.21	(±1.03)	3.11	(±1.12)	0.7297
**Scar**					
Appearance	4.22	(±0.65)	3.99	(±0.67)	0.2183
**Nipple-Areola Complex**					
Nipple symmetry	3.57	(±0.91)	3.46	(±1.06)	0.7416
Nipple position	3.57	(±0.82)	3.65	(±0.91)	0.7484
**Overall Appearance**	**3.58**	**(±0.93)**	**3.38**	**(±1.02)**	**0.4907**

Bold: 3 different aspects are addressed—breast, scar and NAC and finally the overall appearance.

## Data Availability

The data presented in this study are available on request from the corresponding author.

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
