# Peer review of "Long-Term Results after Autologous Breast Reconstruction with DIEP versus PAP Flaps Based on Quality of Life and Aesthetic Outcome Analysis"

_jcm, 2023, doi:10.3390/jcm12030737_

Round 1
Reviewer 1 Report
DIEP flap is considered as the gold-standard for autologous breast reconstruction, whereas PAP flap is an alternative for patients with low BMI and previous abdominal surgery history. There is no need for a direct comparison between the two methods because the indications are different.
Author Response
We highly appreciate the positive assessment of our study! Please see below the response to the critical issues:
Point 1. Reviewer 1.
“DIEP flap is considered as the gold-standard for autologous breast reconstruction, whereas PAP flap is an alternative for patients with low BMI and previous abdominal surgery history. There is no need for a direct comparison between the two methods because the indications are different.”
Response 1. We do agree with you, in terms of considering the DIEP flap as the current gold-standard for autologous breast reconstruction. Still, with ever improving surgical techniques the PAP flap has evolved as a reliable alternative option. Recent studies have proven equivalent microsurgical safety and complication rates of the PAP flap compared to the DIEP flap (Introduction, Line 41-47). This broadens the indication of the PAP flap widely, which is no longer limited to patients with previous abdominal surgery. We observed at our Department that there is a relevant number of patients suitable for both types of reconstruction. Therefore, we are convinced that comparative studies of both techniques are essential in order to facilitate the decision making process and to permit thorough patient counseling. Even more, our data suggest equivalent long-term results concerning quality of life, donor-site morbidity and aesthetic outcome of both flaps. We believe that these results even strengthen our policy of offering both types of reconstruction to eligible patients since the PAP flap has obviously evolved to be on a par with the DIEP flap and therefore, publication of our data will add to the body of literature aiding plastic surgeons in terms of choosing the appropriate reconstructive measures for each individual patient.

Reviewer 2 Report
First of all I would like to express my congratulations to the authors for this well-structured and well-written manuscript. However, there are some minor suggestions to improve this paper:
They conducted a prospective single-center matched cohort study providing and comparing long-term results in 36 patients on patient reported outcome measures as well as donor-site scar assessment and aesthetic evaluation in microvascular DIEP versus PAP flap breast reconstruction. Evaluation included the validated Breast-Q questionnaire for postoperative quality of life and the POSAS questionnaire for evaluation of the donor-site scar quality as well as the Breast Aesthetic Scale for cosmetic analysis. The authors aimed to facilitate the surgeon’s decision-making process thus permitting thorough patient counseling.
However, I was struck that the authors announced the work as the first to compare DIEP and PAP flap long-term results whilst citing Murphy and colleagues, who recently published a comparison of patient reported outcome measures in patients who received DIEP and PAP flap breast reconstructions using the Breast-Q questionnaire (page 2, line 51-54) [Murphy DC, Figus A, Stocco C, Razzano S. A comparison of patient reported outcome measures in patients who received both DIEP flap and PAP flap breast reconstructions. J Plast Reconstr Aesthet Surg. 2019 Apr;72(4):685-710. doi: 10.1016/j.bjps.2019.01.006. Epub 2019 Jan 14. PMID: 30745085.]. Their results suggested that postoperative recovery following PAP flap reconstruction was lighter than after DIEP flap, with a faster return to daily life activities, less pain and a more satisfying donor-site scar, whilst offering equivalent satisfaction with the reconstructed breast.
- I suggest to correct the term “first comparative study” in the introduction section (page 2, line 54) and also in the abstract (page 1, line 12) by deleting the term “first” as the formulation is misleading. Please feel free to correct me, if I am wrong.
- Furthermore, it would be advisable to discuss the results of the above mentioned article in the discussion section. According to Murphy et al. the patients found the PAP flap donor-site more acceptable – please discuss this. They also reported on postoperative pain, which was more bearable two weeks after PAP flap reconstruction. Furthermore, recovery following PAP flap reconstruction seemed less intensive and faster than after DIEP reconstructions. Can you provide data on postoperative pain and recovery of your cohorts? How long did the patients stay in the hospital after surgery? Please state this in the results section.
The investigators included a total of 36 patients after DIEP and PAP flap breast reconstruction in the study. However, one patient was excluded in each group because of flap loss. For reasons of transparency, this should be listed in in the abstract.
In “2.2. Patients” the authors state that they identified all patients with a PAP flap reconstruction from January 2016 and November 2019.
- First question: Can you shortly explain the low number of 18 patients during 46 months? This is less than 3 patients operated on per month. Why didn’t you implement a higher number of patients? Please clarify.
- Second question: It is not clear to me whether this is a prospective study or not. It reads more as a retrospective analysis of patients operated from January 2016 and November 2019. Please clarify.
- Third question: If you collected the data prospectively, why did you match the patients to a retrospective DIEP cohort and did not perform a prospective comparative study?
Lastly, comparing DIEP and PAP flaps in patients with significant differing BMI (25.3 in DIEP flaps and 21.6 in PAP flaps) is in my opinion misleading. In patients with low BMI, harvesting DIEP flaps is limited due to less available volume or may only work out if harvested as a bipedicular DIEP flap. In both cases the DIEP scar is generally very high and thus more visible. Furthermore, patients with a high BMI may quite often benefit from an abdominoplasty and frequently seek the DIEP procedure in order to get rid of their belly fat. That is why I think that the comparison of the DIEP and PAP flap could be futile given the mentioned points.
However, the authors present rare data and their observations may enable us to further improve autologous breast reconstruction. The study is very valuable and might surely support patients and surgeons in the complex decision-making process of microvascular breast reconstruction. I fully agree with the authors that comparative studies regarding the different methods are crucial to support the process of decision-making in patients suitable for both options.
Thank you very much for the opportunity to review this manuscript.
Reviewer 3 Report
Thank you for your interesting submission.
May I ask why you did not use the "secret scar technique" for the pap flap, with the scar hidden in the inguinal crease? Or did you use different flap designs? This might have had an impact on PROM's if you did.
I dont think the conclusions can be drawn as such, because patients chose their donor site probably according to personal preference and abundance of extra fat.
Also I could not see an assessement of PROMs before the surgery. As such it is not clear, if your groups were comparable. Did the patients even fill in the PROMs before their surgery? If not how do you want to assess patient satisfaction without a comparator, or without comparing it to the healthy population.
Round 2
Reviewer 1 Report
The long-term results were well-described.